# Sound and Safe: The Role of Leader Motivating Language and Follower Self-Leadership in Feelings of Psychological Safety

**Milton Mayfield**  **and Jacqueline Mayfield ***

A.R. Sanchez Jr. School of Business, Texas A&M International School, Laredo, TX 780141, USA;
mmayfield@tamiu.edu
* Correspondence: jmayfield@tamiu.edu

**Abstract:** This manuscript presents a study on how leader *motivating language* and follower *self-leadership* act to influence a follower's feelings of *psychological safety*. This study found that both constructs significantly influenced psychological safety in samples from India and the USA. Additionally, this study found that this influence occurred through the mediating processes of *trust in leadership*, *leader inclusiveness*, and *role clarity*. These mediators fully explained motivating language's relationship with psychological safety, but only partially explained self-leadership's relationship. Differences existed in the model between samples, but self-leadership showed an overall consistency between the samples for most relationships. Follow-up analysis indicated that self-leadership without leader communication support lead to a weak or non-existent relationship between self-leadership and psychological safety, but a positive and relatively strong relationship in the presence of motivating language.

**Keywords:** self-leadership; motivating language; psychological safety; leadership; communication



> The ache for home lives in all of us,
> the safe place where we can go as we are and not be questioned.
>
> —Maya Angelou

## 1. Introduction

We all need to feel safe at work and aspire to bring our whole selves there. We want to know that we can be authentic and genuine with our organization's members—to believe that those with whom we spend so much of our lives will support us, and that we can try and fail but still be accepted and respected. If we lose such safety, we tend to withdraw from our work participants—and seek to defend ourselves from the myriad cuts we experience when we fear that they will not protect us or may even threaten us. When we do not feel safe, we eventually stop taking risks including innovating the improvements that organizations need (Mayfield and Mayfield 2012c, 2014a). Put simply, we stop helping those around us to avoid emotional pain, and withdraw from the interpersonal connections at work that should give pleasure to our lives (Edmondson 2018; Edmondson and Lei 2014).

The study of psychological safety focuses a lens on why some work environments promote a sense of inclusion and protection while others lead to a spiral of doubt and insecurity (Edmondson 2018; Edmondson and Lei 2014). More knowledge about how to grow psychological safety can both improve the lives of workers and organizational performance (Appelbaum et al. 2016; Kock et al. 2018). To date, most research on psychological safety has been centered on positive work behaviors at organizational or team levels (Edmondson 2018; Edmondson and Lei 2014). While such interventions promise substantial upgrades in psychological safety, organizations may show resistance to implement these changes on the broader organizational or even team platforms (Cascio 2000; Mayfield et al. 2008). However, changes in leader–follower dyadic behavior—especially through

leader communication such as motivating language (Banks 2014; Brannon 2011)—hold the potential to cultivate psychological safety without requiring extensive organizational shifts (Mayfield et al. 2020; Mayfield and Mayfield 2017b). Going further, an individual leader can implement such changes communicatively in a dyad even when an organization does not recognize the value in improving psychological safety (Ashauer and Macan 2013; Mayfield et al. forthcoming a).

Regrettably, scholars have not fully explored this resource to date. Scholars have made some progress in the links between leader communication and psychological safety by identifying messages from bosses that elicit follower voice and inclusion (Liang et al. 2012; Morrison et al. 2015; Weiss et al. 2018)

But psychological safety can certainly arise from other resources apart from a leader's influence on a follower. He or she can gain a sense of security from establishing a personal level of competence and goals (Godwin et al. 2016; Lovelace et al. 2007). Strong skills in self-leadership can create a internal buffer against workplace stress by developing a sense of self-efficacy (Neck 1996; Prussia et al. 1998). Further, pragmatically, self-leadership is associated with higher job performance which creates external work buffers against many adverse situations (Godwin et al. 1999; Hardy 2004).

Leader motivating language and follower self-leadership can potentially reinforce each other to nurture psychological safety (Manz 1986; Mayfield 2009; Mayfield and Mayfield 2018f; Neck et al. 2016). Here is our study's key contribution: to investigate this possible benefit along with its relevant processes. Just how do we propose this dynamic relationship works? Leaders can access motivating language to support a follower's emotional needs, experienced inclusion and sense of purpose, and guide her/him to optimize performance. In tandem, through tending self-leadership strategies, followers augment the skills and resilience needed to better enjoy a feeling of psychological safety.

This study will examine the effect of leader communication—as captured through the motivating language construct (Mayfield and Mayfield 2018f)—and follower self-leadership skills on psychological safety. Our investigation will explore these relationships by first presenting an overview of psychological safety, then a module about motivating language, and next a review of the self-leadership literature. We then discuss how motivating language and self-leadership should influence psychological safety, followed by our research model and the companion methods section. After model testing, we present this study's contributions and implications for research and practice along with limitations and how these can be addressed.

### 1.1. Psychological Safety

Edmondson (1999) championed the idea of psychological safety—the degree to which an employee feels accepted by and comfortable being open and vulnerable with others at work. She originally conceived psychological safety as a key ingredient for successful team performance since co-operative performance relies on the trust, openness, and bonding that only occurs when someone feels secure and authentic with her or his colleagues (Edmondson 2003; Edmondson et al. 2016). From this initial model, researchers have expanded psychological safety to encompass situations at all research levels: individual to organizational (Edmondson and Lei 2014; Newman et al. 2017). In addition, scholarship has demonstrated that psychological safety plays a crucial role in a myriad of organizational outcomes, including performance, turnover, employee voice, absenteeism, and loyalty (Edmondson 2018; Newman et al. 2017).

At its heart, psychological safety rests on a simple premise: people contribute their best when they feel protected in taking initiatives and expressing themselves at work (Aranzamendez et al. 2015; Idris et al. 2012). When people expect that presenting a new idea will meet with derision, they will not present new ideas. When they anticipate that asking for help will lead to being perceived as weak, they will not ask for help. When they expect that offering assistance will lead to being taken advantage of, they will not offer help. Such an environment also reduces worker motivation and gives cues to expend only

the effort necessary to continue employment and pay (Frazier et al. 2017; Newman et al. 2017).

As Edmondson points out in her original work (Edmondson 1999), this negative situation becomes compounded as the interdependence between workers increases. While reliance on others has always been an organizational characteristic, modern organizations have become increasingly dependent on co-operation between members (Mayfield and Mayfield 2014b, 2019a). Additionally, most high-performing workplaces today demand interdependence, creativity, and flexibility from their members—especially in contrast to the past when the success of most organizations relied on long linked technologies.

Extensive research on psychological safety has shown it to rest on four major antecedents. The first antecedent organizations can do little about—the worker's personality. Fortunately, even the most influential personality trait (having a proactive personality) has a smaller influence (accounting for less than 13% of the variance) on psychological safety than the other three major antecedents: antecedents over which organizations and managers have much greater control (Frazier et al. 2017). For example, a supportive work context explains nearly 25% of the variance in psychological safety. Related to and aiding in the development of a supportive work context, studies have found that work design characteristics account for 28% of the variance in psychological safety and that leader relations account for nearly 20% of the variance (Frazier et al. 2017).

This research indicates that much of how safe a person feels at work rests on how a leader shapes the work environment. However, what this research has overlooked has been a worker's purposeful self-direction (such as self-leadership). Without examining both of these factors in tandem, our understanding of the forces shaping someone's feeling of psychological safety in the workplace will remain incomplete (Edmondson 2018; Mayfield and Mayfield 2018c).

Therefore, we should look at ways that leaders can improve feelings of psychological safety and how follower's self-direction can enhance this state. In our next section, we will examine the leader communication framework motivating language. With this theory, leaders can generate better follower relationships to promote psychological safety and also shape the work environment in ways that will enhance this state.

### 1.2. Motivating Language

Sullivan (1988) put forth motivating language's initial general framework (under the name of motivational language), and other researchers have refined and expanded the idea (Mayfield 1993; Mayfield and Mayfield 2018f). Studies have shown that leader motivating language (ML) significantly and positively influences many critical follower behaviors (including performance, absenteeism, job satisfaction, and effective decision-making) as or more strongly than most other management concepts (Holmes 2012; Mayfield and Mayfield 2018c). Furthermore, investigations have supported ML's generalizability through multiple scholars' congruent findings across multiple settings (Madlock and Sexton 2015; Mayfield and Mayfield 2018b). Of note, causal inferences between motivating language and outcomes have emerged from a quasi-experimental design study (Mayfield and Mayfield 2018f; Wang et al. 2009). As for the structure of motivating language, this conceptualization of positive leader communication categorizes all leader-to-follower oral speech into one of three factors: direction-giving language, empathetic language, and meaning-making language (Mayfield and Mayfield 1995, 2014a, 2019b).

Leaders employ direction-giving language to provide workers with information on requisite actions towards goal attainment, to dispel role ambiguity, and to articulate reward contingencies (Gutierrez-Wirsching et al. 2015; Mayfield and Mayfield 2014b, 2017a). Goal setting and constructive performance feedback are examples of such talk. The second factor, empathetic language, initiates and maintains supportive emotional relations between a worker and a leader (Mayfield and Mayfield 2010, 2016a). Leaders use empathetic language when they praise a worker's successes or advocate for her or his efforts. The third factor, meaning-making language, intersects a follower's personal goals with the organization's

vision, recognizes a follower's unique work contributions, and promotes understanding of the culture (Mayfield and Mayfield 2006, 2009a, 2017c).

Research on motivating language has shown it to have a positive effect on many aspects of a follower's work environment (Mayfield and Mayfield 2004, 2009b, 2018i). Most relevant for this study, several tests have shown motivating language to have a 0.352 correlation with follower job satisfaction—a construct related to but distinct from psychological safety (Mayfield and Mayfield 2018f, 2018j). While not direct evidence for motivating language's effect on psychological safety, these findings give hints that motivating language influences psychological states with a close relation to the construct (Mayfield et al. 1998; Mayfield and Mayfield 2002, 2018k, 2018l). Similarly, research has shown that motivating language has positive relationships with creative support, intrinsic motivation, and satisfaction with a supervisor—all positive work environment factors that should enhance a follower's feelings of psychological safety (Mayfield et al. 1995, 2015). Motivating language has also been positively linked to other aspects of the workplace that research has shown to affect psychological safety (Mayfield and Mayfield 2007, 2012a). Later in this manuscript, we will present further details on how we expect a leader's motivating language to influence psychological safety. However, for the sake of clarity, we will present the study's main hypothesis about the relationship between motivating language and psychological safety.

**Hypothesis 1 (H1).** *Motivating language will have a positive relationship with follower psychological safety.*

### 1.3. Self-Leadership

Self-leadership theory, as pioneered by Manz (Manz 1986; Manz and Sims 1980) and developed by Manz and other scholars (Neck et al. 2016; Neck and Manz 1996), provides a framework for understanding how workers can set goals, develop motivational structures, and develop feedback mechanisms that enhance their personal productivity and well-being (D'Intino et al. 2007; Neck et al. 2004).

Manz developed this model at a time when management research had begun to explore a broader view of leadership—a shift away from the romantic/heroic vision of the leader (Clifton 2019; Du-Babcock and Tanaka 2017). Instead, scholars had begun to examine how followers shape organizations and chart their own courses in work practices. In this atmosphere, self-leadership provides a theory where followers act as their own leaders; basing their actions on internal maps of the world and personal belief systems (Houghton and Neck 2002; Neck et al. 2006). This theory drew from self-management theory (Frayne and Latham 1987; Manz and Sims 1980), but also incorporated elements of intrinsic motivation (Cerasoli et al. 2014; Gerhart and Fang 2015), social learning theory (Frayne and Latham 1987; Manz and Sims 1980), and self-control systems (Forte 2005; Lovelace et al. 2007). In addition to these new elements, self-leadership also included the idea that a follower's purposes could differ from an organization's goals (D'Intino et al. 2007; Neck et al. 2016).

While researchers agree on a core definition of self-leadership, different studies have employed variations of this core (Georgianna 2007; Stewart et al. 2019). As such, this study will adopt the conception used by Houghton, Neck, and colleagues (Houghton et al. 2012; Houghton and Neck 2002). Many self-leadership researchers have employed this conceptualization, and it has provided a strong basis for the well regarded Revised Self-Leadership Questionnaire (RSLQ). From this definition, we will use a conception of self-leadership as a person's ability to motivate themselves in accomplishing tasks they deem relevant. People accomplish this task through three main strategies: behavior focused, constructive thoughts, and natural rewards (Houghton et al. 2012; Houghton and Neck 2002).

People use behavior focused strategies to create rewards linked to goal accomplishment and perseverance in a given task. For example, someone can employ self-cues to

create an enjoyable work environment such as playing enjoyable music or hanging motivational posters. This aspect of self-leadership also encapsulates self-goal setting and the (self-administered) rewards someone sets for achieving these goals. Engaging in this form of self-leadership also helps people with self-understanding through their actions (Houghton et al. 2012; Houghton and Neck 2002; Neck and Manz 2012; Seligman and Csikszentmihalyi 2014).

Self-leadership also involves constructive thought strategies. These strategies use positive self-talk, mental rehearsing, and conscious evaluation of personal beliefs and values. This activity involves the mental rehearsal and visualization of a successful task performance, engaging in in positive self-talk, and an evaluation and articulation of beliefs and assumptions (Neck and Houghton 2006; Houghton and Neck 2002; Neck and Manz 2012; Stewart et al. 2011).

Finally, people adept in self-leadership employ natural rewards (Houghton and Neck 2002). For natural rewards, people search out ways to make a task pleasant in-and-of itself or the positive aspects of a task. People can do this by creating a positive environment to complete a task, such as writing a report in an outdoor setting or while listening to pleasing music. People can also enhance natural rewards by reminding themselves of the task's higher purpose, such as keeping in mind that filling out inventory reports helps the organization track needed resources (Neck and Houghton 2006; Manz and Sims 2001; Stewart et al. 2011).

Self-leadership research has shown that it differs from personality (Neck and Houghton 2006; Houghton et al. 2004) and that training (and thus environmental forces) can alter its level (Neck and Manz 1996; Stewart et al. 2011). Such training influences opens up organizational possibilities since researchers have found that self-leadership improves important outcomes including absenteeism, ethical behavior, organizational commitment, organizational citizenship behavior, effective goal setting, creativity and innovation, and team potency and trust (Neck and Houghton 2006; Stewart et al. 2011). In an international context, cross-cultural tests of the RSLQ have been conducted in five different languages, with construct validities that are generally consistent with the original instrument (Houghton et al. 2012).

In addition—and more directly relevant for this research—several studies have shown that self-leadership has a positive relationship with affective states that influence or relate to psychological safety. These constructs include stress reduction, self-efficacy, and psychological empowerment (Neck and Houghton 2006; VanSandt and Neck 2003).

The next section develops how we expect a person's self-leadership ability and their leader's motivating language use to influence psychological safety. Before that section, however, we will present the study's main hypothesis about the relationship between self-leadership and psychological safety.

**Hypothesis 2 (H2).** *Self-leadership will have a positive relationship with follower psychological safety.*

*1.4. The Relationships between Motivating Language and Self-Leadership and Psychological Safety*

This section will examine how we expect motivating language and self-leadership to influence a person's psychological safety. By how, we mean what processes motivating language and self-leadership activate in order to change psychological safety. We have already presented our two main hypothesis about what the relationships should be—that both motivating language and self-leadership should have a positive relationship with self-leadership. Now, we want to explore *why* we believe these relationships exist.

Understanding the processes by which relationships operate helps us to advance a field of study in two ways. First, it gives us a better understanding of the mechanisms by which constructs influence each other. This understanding provides us with ways to develop practical interventions that should operate as intended. Without an understanding

of these mechanisms, interventions may fail mysteriously if outside forces hinder changes in the mediator constructs (Cascio 2000; Pearl 2009).

From a research standpoint, better understanding of the mediating variables in a relationship provides us with many advantages. When we can successfully identify the mediating variables in a relationship, we increase the chances that we have a true understanding of the causal processes in that relationship (Pearl 1998, 2009). The identification of causal mechanisms allows us to see *how* two constructs relate rather than just showing evidence for a relationship, and thus, the identification provides a test of our understanding of the true relationship between constructs. If we find that constructs fully mediate a relationship, then we have evidence of a good understanding of the processes, while partial mediation indicates that we need to investigate the relationship further (Mayfield et al. 2020; Pearl 1998).

Now that we have briefly stated the reasons we want to understand mediating variable, we can discuss how we expect those relationships to operate in our model. For our first statement, we will drawn on prior research and empirical evidence. Prior research has shown that leader motivating language has a significant and positive relationship with follower self-leadership (Mayfield et al. forthcoming a; Mayfield and Mayfield 2016b). As such, we will present our next hypothesis based on findings from this study.

**Hypothesis 3 (H3).** *Leader motivating language will have a positive relationship with follower self-leadership.*

We expect motivating language to influence a follower's psychological safety through three mechanisms: the follower's trust in the leader, how inclusive a follower believes a leader to be, and how clearly a follower understands her or his role. As discussed in the section on psychological safety, research has shown these three mechanisms to have a positive and substantial influence on follower's feelings of psychological safety (Frazier et al. 2017; Newman et al. 2017). For the remainder of this section, we will outline how we expect motivating language to influence these three constructs.

First, motivating language should help develop trust between a leader and a follower (Mayfield and Mayfield 2018f). Direction-giving language—which provides information on performance expectations—helps develop procedural trust in a leader: the follower knows what the leader expects (Mayfield and Mayfield 2018b, 2018d). Empathetic language helps create emotional bonds that should also encourage follower trust (Mayfield and Mayfield 2018g, 2018e). Finally, meaning-making language creates trust by developing shared goals: people have more trust in someone when they both work toward a common end. Hypothesis 4 provides a formal statement of this expectation.

**Hypothesis 4 (H4).** *Motivating language will positively influence a follower's trust in a leader.*

We also expect that self-leadership will influence trust in a leader, but for different reasons. Self-leadership should generate trust in a leader through the follower's own self-efficacy. Self-leadership research has shown positive relationships to such constructs as self-efficacy and self-confidence (Paglis 2010; Prussia et al. 1998). When a person feels greater levels of security in their own abilities, they also tend to feel more comfortable trusting others. This trust comes from a belief that they can more easily deal with the consequences if they make a mistake in trusting someone else.

To explore this idea more, we first need to examine what we mean by trust in a leader. Researchers have divided trust into two parts: affective and cognitive (Baer et al. 2020; Ling and Guo forthcoming). With cognitive trust, someone has confidence that they can predict what a person will do in different circumstances—that they know what to expect whether that behavior is positive or negative (Edmondson and Lei 2014; Schaubroeck et al. 2013). For such trust, we expect that as someone's self leadership increases, they will develop greater capabilities in predicting a leader's behavior and feel more confident in this capability.

Affective trust happens when a person feels that someone else will act in a positive way towards her or him (Edmondson and Lei 2014; Ling and Guo forthcoming). As self-leadership increases, research has shown that leaders act more favorably towards the follower (Neck and Houghton 2006; Stewart et al. 2011), thereby increasing the trust a follower has for the leader.

Based on these ideas, self-leadership should have a positive association with a follower's trust in a leader. Hypothesis 5 formally states this idea.

**Hypothesis 5 (H5).** *Self-leadership will positively influence a follower's trust in a leader.*

We base out next hypothesis on prior research. Several studies have shown that trust in a leader will increase feelings of psychological safety (Roussin 2008; Roussin and Webber 2012). To briefly summarize this line of research, as people trust a leader more, they come to feel more secure in their workplace. This security arises from being able to predict what will occur in a given circumstance (thus reducing uncertainty and role ambiguity), and believe that a leader will look out for a follower's best interests. These factors lead to workers feeling that the workplace provides a safe area for self-expression. Hypothesis 6 provides a formal statement of this idea.

**Hypothesis 6 (H6).** *A follower's trust in a leader will positively influence the follower's feeling of psychological safety.*

Motivating language should positively influence how included a follower feels by her or his leader. Meaning-making language should strongly influence this feeling of inclusiveness. Through meaning-making language, a leader helps a follower see how her or his personal goals align with organizational or work-group goals (Mayfield and Mayfield 2018a, 2018c). This use of leader communication will help the follower understand how the leader takes her or his needs into account when achieving workplace goals. Similarly, empathetic language gives the follower a feeling of bonding with the leader, and direction-giving language can help the follower understand goals—thus creating a feeling of understanding about needed tasks (Mayfield and Mayfield 2018a, 2018f, 2018h). Hypothesis 3 provides a specific statement about how this mechanism should operate.

**Hypothesis 7 (H7).** *Motivating language will positively influence a follower's feeling of a leader's inclusiveness.*

Higher self-leadership should lead to being included more by a leader for several reasons. From a perceptual standpoint, when a person practices the constructive thoughts aspect of self-leadership, he or she will be less likely to interpret events in a negative light (Houghton et al. 2012; Neck et al. 2006). For example, someone high in self-leadership will more likely interpret a boss as being busy during inventory planning time rather than trying to exclude the follower.

In addition to increasing the perception of inclusiveness, self-leadership should also generate behaviors that lead to *actual* increased leader inclusiveness (Neck and Houghton 2006; Godwin et al. 1999). Research on self-leadership has consistently shown a positive link with follower performance and other positive workplace behaviors such as organizational commitment (Konradt et al. 2009; Prussia et al. 1998), and such behaviors should make a leader more open to including a follower in workplace interactions (Mayfield and Mayfield 2009a; Rockstuhl et al. 2012). Finally, self-leadership behaviors also help a follower focus more on an organization's bigger picture—a view broader than just personal concerns as demonstrated through self-leadership's positive relationships with organizational commitment and citizenship behavior (Mansor et al. 2013; Park and Park 2008). With this broader view of the organization, a follower can see a wider scope for what inclusion means, and leaders will have more opportunities to include the follower in activities.

The following hypothesis formally states how we expect these ideas to operate.

**Hypothesis 8 (H8).** *Self-leadership will positively influence a follower's feeling of a leader's inclusiveness.*

As with the prior study mediators, we draw the next hypothesis from existing studies. Research has shown that leader inclusion fosters a sense of psychological safety in followers (Frazier et al. 2017; Newman et al. 2017). Being included fosters this affective state by creating a feeling of being part of the work team or organization (Carmeli et al. 2010; Roussin and Webber 2012). In addition, when people feel included in organizational or team aspects, they will feel more in control of their work situation and thus have greater feelings of safety (Hirak et al. 2012; Nembhard and Edmondson 2006). Our next hypothesis distills these ideas into a formal statement.

**Hypothesis 9 (H9).** *A follower's feeling of being included by a leader will positively influence the follower's feeling of psychological safety.*

Motivating language should influence a follower's role clarity. The direction-giving language construct partly derives from the idea of role clarity and centers around how well a follower understands task requirements (Mayfield and Mayfield 2009b, 2017a). Additionally, while direction-giving language lets the follower know what he or she needs to do, meaning-making language provides the answer to why the task must be completed. These two aspects of motivating language help a follower fully understand what needs accomplishment. Empathetic language, while not expected to play such a direct role, helps followers by providing emotional support when taking on new tasks, thus also helping improve role clarity (Mayfield and Mayfield 2018d, 2018e). Hypothesis 10 provides a testable statement about these ideas.

**Hypothesis 10 (H10).** *Motivating language will positively influence a follower's role clarity.*

Evidence also suggests that self-leadership should increase follower role clarity. Increased self-leadership use will lead people to better define their roles and work tasks. This process occurs through self-leadership's many feedback loop mechanisms such as self-reflection (Neck and Houghton 2006; Stewart et al. 2011). Through such processes, people will make regular adjustments to their performance activities, and thus increase their clarity about such expectations. In addition, as stated in earlier discussions about self-leadership behaviors, people who practice self-leadership more will receiver greater attention and feedback from their superiors. This feedback will also increase role clarity. Hypothesis 11 provides a formal statement of these ideas.

**Hypothesis 11 (H11).** *Self-leadership will positively influence a follower's role clarity.*

For our final hypothesis, we will examine the link between role clarity and psychological safety. Role clarity should have a positive link with psychological safety by reducing ambiguity and increasing self-efficacy (House and Rizzo 1972; Paglis 2010). With reduced ambiguity, a worker will have less stress and feel greater security in their job tasks. This more relaxed feeling at work will foster an atmosphere that promotes psychological safety (Dollard et al. 2012; Frazier et al. 2017). In addition, increased role clarity will generate greater self-efficacy in job tasks. Similarly, as a worker's self-efficacy increases, her or his stress should decrease. Additionally, increased self-efficacy will engender a feeling of confidence about handling workplace activities (Bandura 1977; Gist and Mitchell 1992). In total, these effects should lead to increased feelings of psychological safety, as formally stated in Hypothesis 12.

**Hypothesis 12 (H12).** *A follower's role clarity will positively influence the follower's feeling of psychological safety.*

*1.5. Research Questions about Mediation and Generalizability*

While we expect the three constructs discussed above to mediate the relationship between motivating language and self-leadership and psychological safety, prior research provides less guidance on if the constructs will fully or partially account for the relationship. As such, we plan to explore this relationship through a research question.

RQ$_1$: Do the constructs of trust in leadership, leadership inclusiveness, and role clarity fully or partially mediate the relationship between leader motivating language and follower psychological safety?

RQ$_2$: Do the constructs of trust in leadership, leadership inclusiveness, and role clarity fully or partially mediate the relationship between worker self-leadership and psychological safety?

We also plan to explore whether this relationship can generalize outside of the USA. Prior motivating language (Madlock and Hildebrand Clubbs forthcoming; Mayfield and Mayfield 2018c) and self-leadership (Georgianna 2007; Ram 2015) research has shown the construct's relationships to operate across cultural boundaries (Mayfield and Mayfield 2018f), so we expect our model to generalize as well. However, finding differences could provide greater insights into how these constructs operate in different cultures (House et al. 2004). Our second research question provides a statement of what we will look for.

RQ$_3$: Does the model operate in the same way outside in India and the USA?

Figure 1 provides a graphical representation of our model.

This figure provides a graphical depiction of how motivating language and self-leadership should affect follower psychological safety. The top—smaller—depiction shows the expected direct, positive influence of motivating language and self-leadership on follower psychological safety (and the expected effect of motivating language on self-leadership). The lower—larger—depiction shows the expected mechanisms by which motivating language and self-leadership influences psychological safety. These influences—*trust in leadership*, *leadership inclusiveness*, and *role clarity*—should mediate between motivating language and follower psychological safety and self-leadership and psychological safety. These mediators translate leader communication and follower skills into follower congnitive states that promote a feeling of psychological safety. The model proposes a positive relationship between all constructs. The dotted line represents how national context might lead to different strengths of relationships between constructs.

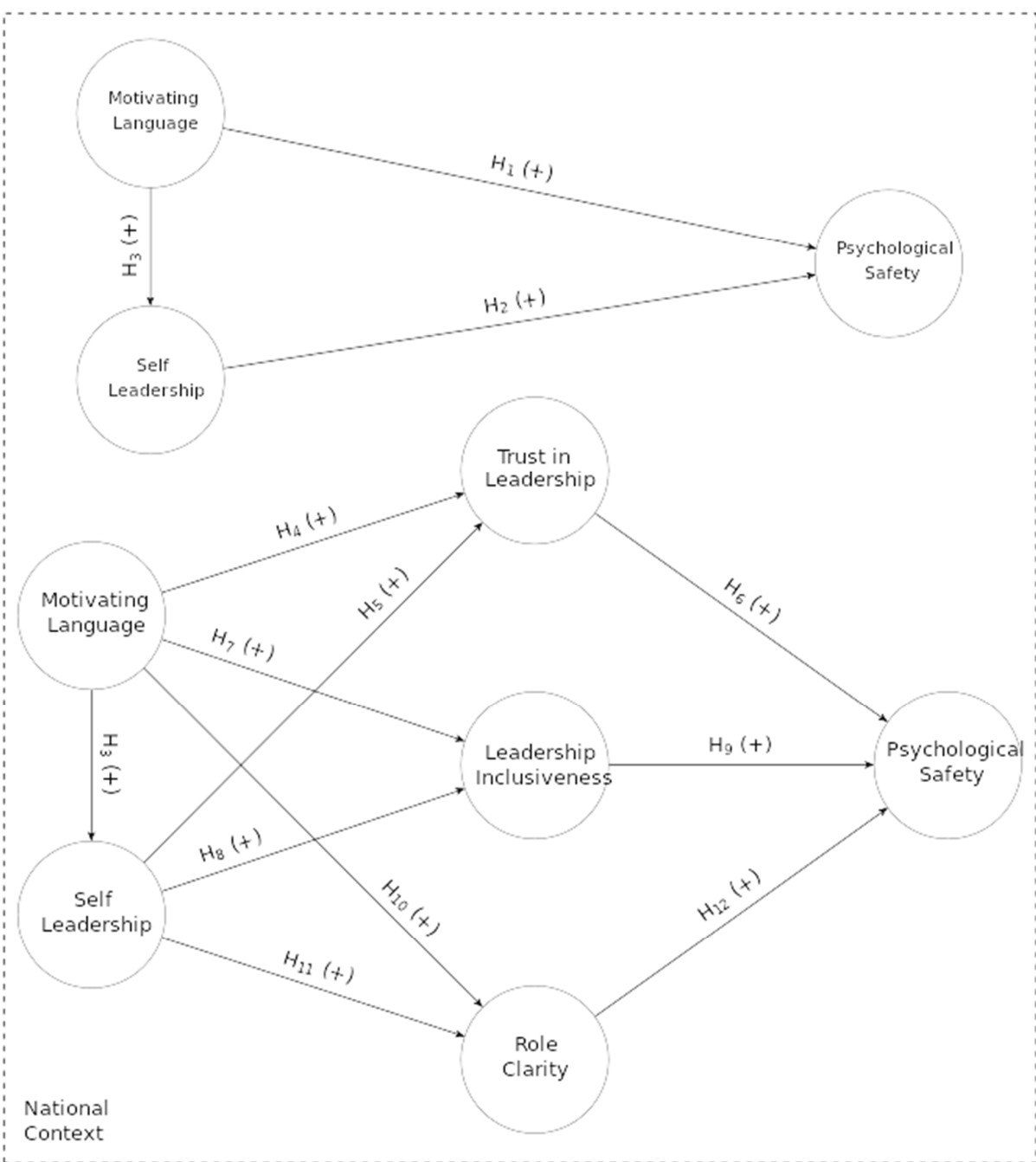

**Figure 1.** A proposed model of how leader motivating language and follower self-leadership influences follower psychological safety.

## 2. Results

### 2.1. Methods

#### 2.1.1. Sample

To test our model, we obtained samples from India and the USA. These samples provide us with a method to compare our model across highly divergent settings—both in terms of cultural and economic environments (Dorfman et al. 2004; Mayfield and Mayfield 2012b). It was beyond this study's scope to examine the effects of specific differences between these settings, but using such varied settings permits us to examine the settings' effects as a whole and our model's generalizability (Cronbach et al. 1963; Mayfield and Mayfield 2018b).

We used Amazon's Mechanical Turk service to collect our data (Buhrmester et al. 2011; Crump et al. 2013). Mechanical Turk gives researchers an avenue to post calls for survey completion in return for a specified fee (Difallah et al. 2018; Huff 2014). Research on the quality of these responses have consistently shown that Mechanical Turk participants provide responses at least as good at those collected through traditional means (Crump et al. 2013; Kees et al. 2017). In addition, this service allows data collection from a broad cross-section of respondents—a characteristic often lacking in traditional samples (Hauser and Schwarz 2016; Kees et al. 2017).

To ensure quality, we followed generally accepted guidelines for improving survey responses (Berinsky et al. 2014; Dillman et al. 2014). First, we provided a payment slightly above the average for surveys on Mechanical Turk (US$0.70 per respondent for a completed survey). We also used well-tested measures (or adaptions of such surveys). In addition, we ensured respondent anonymity and stressed that the responses were only for research purposes. Finally, we included two attention check items to catch respondents answering in a haphazard fashion (Berinsky et al. 2014; Mayfield et al. forthcoming b).

We requested 400 respondents from India and from the USA. We had more respondents than requested (433 from India and 473 from the USA). This excess of respondents came from people who did not request payment since the job posting remained open until 400 respondents collected payment in each country.

We selected these nations for the insights we hoped they could provide into motivating language and self-leadership. The USA has seen substantial research on the two focal constructs (Mayfield and Mayfield 2018f; Neck et al. 2006), and a sample from this nation provides a base to compare results from other nations. In contrast, researchers have conducted fewer investigations on either construct in India, but this nation provides a good potential contrast for findings from the USA. India possesses a culture with markedly different cultural characteristics, demographic aspects, and even predominate religious beliefs from the USA. Therefore, findings of similar results across these two nations provide good evidence of the model's generalizability, while differences can provide avenues for future researchers to explore to try and identify why the differences exist (i.e., which national characteristics create the differences).

After we collected the data, we removed the respondents who did not answer both attention check items correctly. The attention checks were two items placed in two different scales. Each item asked the subjects to select a specific response: one required respondents to select "somewhat agree" and the other "somewhat disagree." By using these attention check items, we identified respondents who either carelessly responded, selected items at random, or answered using the same response for each question. After removing careless responders, we had 427 respondents from India and 452 respondents from the USA.

We also used randomization to reduce systematic order bias in responses (Abraham et al. 2009; Dillman et al. 2014). Each scale appeared in a random order for all respondents, and all items within in a scale appeared in a random order for each respondent.

### 2.1.2. Sample Demographics

The study sample contained a diverse cross-section of respondent backgrounds. In the USA, the respondents had a mean age of 39.46 years, and 49% of respondents were female. Respondents had an average of 15.3 years of full-time work experience, had worked for their employer for 7.16 years, been in the same position for 5.95 years, and with their current supervisor for 4.87 years. In addition, respondents came from a wide variety of industries and organizational positions, and mirrored general USA workplace demographics.

The Indian respondents were younger (being an average age of 30.49 years old), and the sample had a smaller proportion of female respondents (with only 33% female). The respondents had an average of 6.74 years full-time work experience. They had worked for their current employer for 5.42 years, been in their current position for 4.13 years, and with their current boss for an average of 4.23 years (the discrepancy of being with a boss longer than in a position came from missing data from some respondents). The Indian sample had

respondents from a wide variety of industries, but showed a higher concentration from the IT industry than general national demographics.

*2.2. Measures*

In measuring our constructs, we used well-established scales—either in their original form or slightly adapted to fit our hypotheses. To measure leader motivating language, we employed Mayfield and Mayfield's (Mayfield et al. 1995; Mayfield and Mayfield 2017a) Motivating Language scale. Similarly, we used Houghton and Neck's Revised Self-Leaderhsip scale (Houghton and Neck 2002) to measure a respondent's self-leadership. To capture a respondent's trust in their leader, we used the supervisory section of Nyhan and Marlowe's (Nyhan and Marlowe 1997) Organizational Trust scale. We used Carameli, Reiter-Palmon, and Ziv (Carmeli et al. 2010) Inclusive Leadership scale to measure leader inclusiveness. For role clarity, we used Rizzo, House, and Lirtzman's (Rizzo et al. 1970) scale. Finally, we adapted Amy Edmondson's (Edmondson 1999, 2018) Team Psychological Safety measure to capture how safe someone felt with their leader. To adapt this scale, we changed to focal wording in items from team to leader. For example, where the original scale had "If you make a mistake on this team, it is often held against you", we used "If you make a mistake, your boss will hold it against you." We provide examples from these scale in Appendix A.

All study scales demonstrated acceptable reliabilities as measured through Cronbach's alpha and the G6 reliability measure. Table 1 presents the reliabilities of these measures. For the motivating language and self-leadership scales, the table presents both the subscale and overall scale reliabilities.

**Table 1.** Study Measure Reliabilities.

| Measure | Alpha | G6 |
|---|---|---|
| Direction-Giving | 0.92 | 0.91 |
| Empathetic | 0.88 | 0.87 |
| Meaning-Making | 0.90 | 0.90 |
| Motivating Language, overall | 0.91 | 0.88 |
| Self-Leadership, Behavior Focused Strategies | 0.81 | 0.79 |
| Self-Leadership, Natural Rewards | 0.76 | 0.72 |
| Self-Leadership, Constructive Thoughts | 0.80 | 0.74 |
| Self-Leadership, overall | 0.91 | 0.92 |
| Psychological Safety, Leader | 0.70 | 0.75 |
| Leadership Inclusiveness | 0.92 | 0.91 |
| Trust in Supervisor | 0.93 | 0.92 |
| Role Clarity | 0.75 | 0.71 |

Table 2 presents construct descriptives.

**Table 2.** Study Measure Descriptives.

| Construct | Mean | Std. Dev. |
|---|---|---|
| Motivating Language | 3.41 | 0.78 |
| Self-Leadership | 3.67 | 0.58 |
| Trust in Supervisor | 5.15 | 1.03 |
| Leader Inclusiveness | 3.71 | 0.83 |
| Role Clarity | 4.08 | 0.57 |
| Psychological Safety | 3.45 | 0.70 |

As an initial data check, we examined the raw correlations between our constructs. All correlations were significant and in the expected (positive) direction. In addition, all relationships appear substantially linear. Table 3 presents construct correlations.

**Table 3.** Study Variable Correlations.

| Motivating Language | Self-Leadership | Trust-in-Supervisor | Leader Inclusiveness | Role Clarity | Psychological Safety |
|---|---|---|---|---|---|
| Motivating Language | 0.58 | 0.42 | 0.48 | 0.39 | 0.23 |
| Self-Leadership | | 0.40 | 0.38 | 0.47 | 0.10 |
| Trust-In-Supervisor | | | 0.70 | 0.60 | 0.60 |
| Leader Inclusiveness | | | | 0.54 | 0.61 |
| Role Clarity | | | | | 0.43 |
| Psychological Safety | | | | | |

In addition to the steps we took to ensure quality scale responses, we also wanted to control for possible common methods bias (Podsakoff et al. 2003, 2012) and social response bias (Dillman et al. 2014; Graeff 2005) in this study. To do so, we used the technique of measurement and control. We included two scales to measure possible response bias—the Marlowe–Crowne (Reynolds 1982) Social Response scale, and Mayfield's (Mayfield et al. forthcoming b) Comparative Taste Preference scale. The Social Response scale captured how likely someone was to answer a question based on how they felt they should answer rather than her or his actual situation.

We used the Comparative Taste Preference scale to capture common methods bias. Common methods bias can occur whenever a respondent answer questions in a systematic way because of forces outside of the measured constructs (Podsakoff et al. 2012; Spector 2006). For example, a person might tend to answer toward the extremes of a scale, or an unhappy mood at the time of response might influence all of the respondent's answers in a given direction. The presence of common methods bias can either inflate or reduce apparent correlations between constructs, thus leaving the actual relationships in question (Mayfield et al. forthcoming b; Podsakoff et al. 2003).

To test for and deal with common methods bias, you can include a wholly unrelated construct in a questionnaire. To do so, you include some construct that should have no relationship with other constructs, but use the same type of measurement method as other items in the questionnaire. Therefore, any relationship between the unrelated construct and the study constructs must result from common methods bias.

We used the social desirability and unrelated construct measures as predictors for our model constructs in a multivariate regression analysis. This analysis showed significant (through relatively weak) relationships between the predictors and dependent variables. As such, this test indicated the presence of response bias among respondents.

To remove this bias, we took the construct residuals from our bias analysis, and used these residuals in our full analysis. Using the residuals gives us a data set that has had the tested biases removed from it.

We also compared the correlations from the raw data set and the adjusted data set. Examining the difference between these two correlation matrices showed that the response bias only created minor *inflation* in relationships. The highest increase in relationships came with the motivating language and self-leadership constructs, where response bias increase the apparent relationship by 0.02 points.

However, response bias did seem to have a stronger effect in *depressing* the relationship between motivating language and psychological safety (by 0.10 points), and self-leadership and psychological safety (0.06 points).

We present the corrected correlation matrix in Table 4 below.

**Table 4.** Study Variable Correlations, Controlled for Bias.

| Motivating Language | 0.56 | 0.46 | 0.51 | 0.39 | 0.34 |
|---|---|---|---|---|---|
| Self-Leadership | | 0.43 | 0.40 | 0.48 | 0.16 |
| Trust-In-Supervisor | | | 0.69 | 0.60 | 0.59 |
| Leader Inclusiveness | | | | 0.53 | 0.61 |
| Leader Inclusiveness | | | | | 0.44 |
| Psychological Safety | | | | | |

The Comparative Taste Preference scale had a reliability of 0.87 in this study. The Social Desirability Response scale had a lower reliability of 0.58, but this alpha comes from the scales yes/no response format, and is in line with most reported uses of the scale (Reynolds 1982; Richman et al. 1999).

You can find copies of the Motivating Language and Comparative Taste Preference scales in Appendix A. The original authors released the scales under a Creative Commons license allowing reproduction. We cannot reproduce the other scales for copyright reasons, but the interested reader can find full copies in the cited publications.

*2.3. Results*

We used path analysis to test our overall model. With path analysis, a researcher can examine complex, mediated relationships. This technique has also proven robust against assumption violations that can distort results from other complex model tests such as structural equation modeling and PLS (Chin 1998; Lleras 2005). In addition, a path analysis has substantially greater power than an equivalent latent variable method (Cohen 1988; Lleras 2005). We used the lavaan package (Rosseel 2012) for the statistical software R (R Core Team 2020) for our analysis.

We selected lavaan and R for our analysis software because of their flexibility, substantial use in the research community, and the quality assurance benefits attendant with open source software (Muenchen 2012; Zhao and Elbaum 2003). With its integration into R, researchers can seamlessly integrate data testing (as for reliability), manipulations (as with the removing bias stage in this research), and visualizations (as with the correlations matrices). This workflow ease allows for greater focus on interpreting the results. In addition, R software has become the second most cited software among academic scholars (Muenchen 2012).

For the constructs, we averaged all items to create variables for the model. For motivating language and self-leadership, we first averaged items in each measures subscales (e.g. direction-giving language or natural rewards), and then took the averages of these subscales to create the overall score for each variable.

As a first step in our analysis, we checked to see if the model operated the same across our two samples. To do so, we constrained all paths in our model to have the equal values across the Indian and USA groups. This analysis was equivalent of having both set of subjects included in an analysis with no distinction between the groups. Next, we constructed a model where all paths could vary between samples. This analysis was the same as if we had analyzed each group separately. As a final step, we compared these two models to see if they had significant differences in these path values using an Analysis of Variance method (Bollen and Long 1993; Mayfield et al. forthcoming b). From this comparison, we saw that the two models significantly varied across at least one path.

To determine which paths differed significantly, we created a model that allowed only one path to vary between the models—the path with the least difference between the two unconstrained models. We then checked if this model showed significant differences from the completely unconstrained model (the model with all paths allowed to vary). When we found significant differences between the two models, we continued to free paths in the same way until we no longer found differences.

For the final model, we had to free all but the following paths: motivating language —>self-leadership, self-leadership—>psychological safety, trust-in-leadership—>psychological safety, and role clarity—>psychological safety.

To test whether our mediating variables fully or partially accounted for the relationship between motivating language and psychological safety, we created a direct path between the two variables. The path was non-significant, indicating that the three mediating constructs fully accounted for motivating language's influence on a follower's feeling of psychological safety. As such, these three constructs fully mediate the relationship between motivating language and psychological safety.

We followed the same process to test for the relationship between self-leadership and psychological safety. This analysis showed a significant, direct relationship between the constructs. This result showed that the constructs only partially mediated the relationship between self-leadership and psychological safety. This finding indicates that some other mechanism operates to transmit self-leadership's influence to psychological safety in addition to those tested. Figure 2 provides a graphical presentation of our model results.

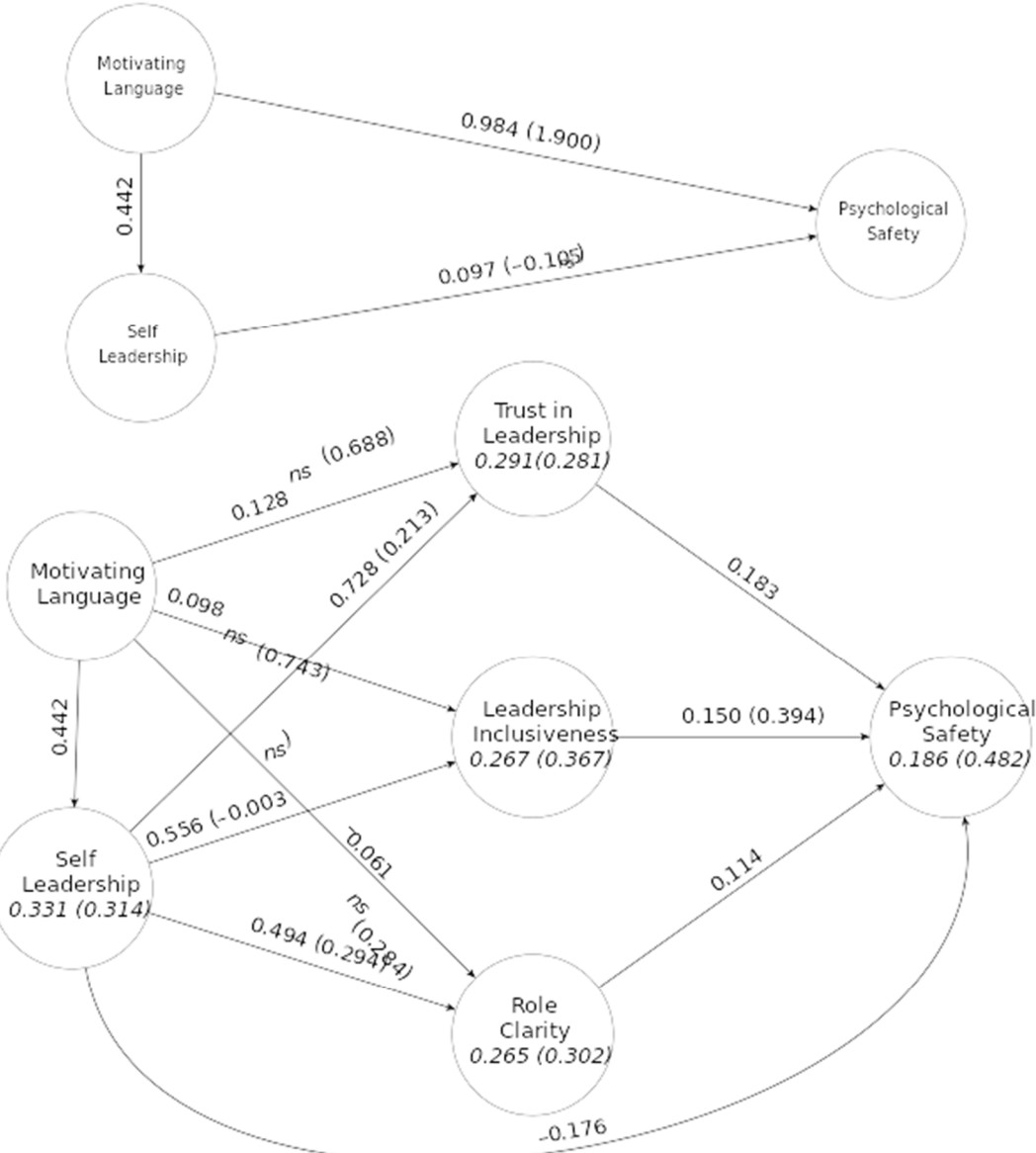

**Figure 2.** Model results for leader motivating language and follower self-leadership influences on follower psychological safety.

This figure shows the model results for how leader motivating language and follower self-leadership influence follower psychological safety. When paths differed significantly between India and the USA, they were presented in standard format (for India) and in parentheses (for the USA). The model denotes non-significant relationships with an ns after the path coefficient. The $R^2$ for each construct appears in italics below the construct name. The top – smaller – depiction shows the total effects that motivating language has on self-leadership and psychological safety, and the total effect that self-leadership has on psychological safety. The lower—larger—depiction shows the full model relationship. The model shows a fully mediated relationship between motivating language and psychological

safety, and a partially mediated relationship between self-leadership and psychological safety.

The model shows that the relationship between motivating language and self-leadership remains the same across the two samples (0.442), although the variance accounted for differs slightly between these nations ($R^2$ of 0.331 in India, and 0.314 in the USA). Where paths differ between nations, motivating language has a stronger influence in the USA (with no significant direct effect on any of the mediating variables), and self-leadership has a stronger effect in India. These results mean that, in India, motivating language effects psychological safety through a double mediating process: motivating language—>self-leadership—>other mediating variables—>psychological safety.

This model also showed that leader motivating language has a very strong influence on psychological safety. In India, motivating language has a standardized total effect of 0.984. In the USA, it has a standardized total effect of 1.900. Standardized paths express relationships in terms of standard deviations. This means that in the USA, for every one standard deviation increase in leader use of motivating language, you can expect to see nearly two standard deviation increases in psychological safety. In India, you can expect to see nearly a one-to-one relationship between the two constructs.

We have deferred discussing self-leadership in this model because of an oddity in our results. The reader may have noticed some discrepancies between construct correlations and the path analysis. In the correlations matrix, self-leadership demonstrates significant, positive, and relatively strong relationships with all other model constructs. However, in the model many of these paths become non-significant.

The reason for these changes comes from the high collinearity between motivating language and self-leadership. Path analysis uses regression techniques as its base. As such, when two correlated constructs predict a set of endogenous variables, the analysis methods partials the shared variance between the two correlated variables. Therefore, one variable will act as it was originally conceived and measured, while the other acts as the original construct controlled for another construct (Cohen et al. 2003; Voss 2005).

With an idea of how correlated exogenous variable act in a path analysis, we can now take a deeper look at why self-leadership may behave in a counter-intuitive way in the model, and develop a strategy to deal with this behavior. For this examination, we need to realize that the analysis construct does not measure self-leadership, but rather self-leadership controlled for leader motivating language. In other words, the construct measures how a follower's self-leadership would act in the absence of leader motivating language use.

While we cannot know without further investigations what this analytic constructs truly looks like, we can broadly paint it as self-leadership without leader guidance or support (as expressed through leader verbal behaviors). We may consider this construct as a kind of insular self-direction and motivation where someone would have to set their own goals and rewards through a mechanism of using environmental cues about what these goals should be, and for what behaviors they should create a reward system. With this type of self-leadership, we can better understand why its link with psychological safety would prove weak.

However, while this insular version of self-leadership may provide grounds for future research, we care more about the traditional view of self-leadership in this study. Pearl (2009, 2016) has shown that when a set of variables fully mediates the relationship between two variables, you can drop the fully mediated variable with no explanatory loss to the (terminal) endogenous variable. As such, we can remove motivating language from the model and see self-leadership's full influence.

To do so, we re-ran the model without motivating language. We used the same process to compare results between nations as in the original model, although we found that only the leadership inclusiveness—>psychological safety link differed significantly between the two nations. As such, self-leadership seems to have a more generalizable relationship with the other constructs than motivating language does.

For our broader results, self-leadership had a stronger and positive total relationship with psychological safety, and far stronger relationships with the mediating variables than the model including motivating language. Self-leadership still have a negative direct (non-mediated) relationship with psychological safety, but this relationship arises from the same multi-collinearity issues we saw with motivating language, only with the mediating variables. While researchers may want to investigate this negative direct relationship, for this study, this second model provides an adequate answer to our question of how self-leadership relates to psychological safety.

Figure 3 provides a graphical representation of this second model.

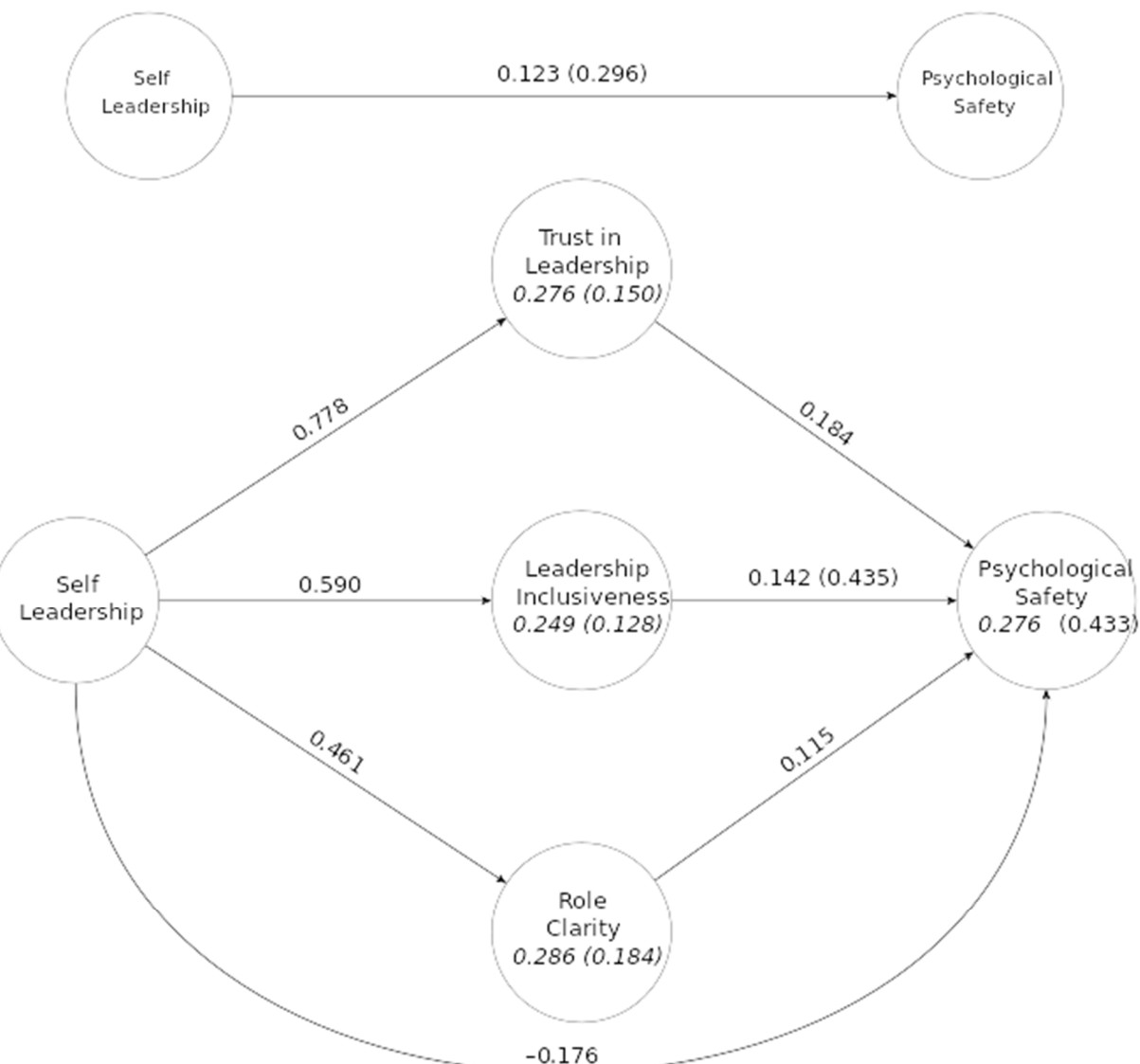

**Figure 3.** Model results for follower self-leadership influences on follower psychological safety.

This figure shows the model results for how follower self-leadership influences follower psychological safety. When paths differed significantly between India and the USA, they were presented in standard format (for India) and in parentheses (for the USA). The $R^2$ for each construct appears in italics below the construct name. The top—smaller—depiction shows the total effects that self-leadership has on psychological safety. The lower—larger—depiction shows the full model relationship. The model shows a partially mediated relationship between self-leadership and psychological safety.

We also present our results in tabular form below. The Table 5 shows the direct, indirect, and total effects of all variables. Results in bold show the paths for the USA sample when significant differences exist between India and the USA. Paths in italics indicate a lack of difference between the samples. Table 6 shows results for the model without motivating language.

**Table 5.** Full Model.

| From | To | Direct Effects | Indirect Effects | Total Effects |
|---|---|---|---|---|
| Motivating Language | Psychological Safety | $0.005_{ns}$ <br> **$0.093_{ns}$** | 0.979 <br> **1.807** | 0.984 <br> **1.900** |
| Self-Leadership | Psychological Safety | $-0.176$ | 0.273 <br> $0.071_{ns}$ | 0.097 <br> $-0.105_{ns}$ |
| Motivating Language | Self-Leadership | *0.442* | *0.442* | *0.442* |
| Motivating Language | Trust in Leadership | $0.128_{ns}$ <br> **0.668** | 0.322 <br> **0.094** | 0.450 <br> **0.782** |
| Self-Leadership | Trust in Leadership | 0.728 <br> **0.213** | NA | 0.728 <br> **0.213** |
| Trust in Leadership | Psychological Safety | *0.183* | NA | *0.183* |
| Motivation Language | Leadership Inclusiveness | $0.098_{ns}$ <br> **0.743** | 0.246 <br> **$-0.002_{ns}$** | 0.344 <br> **0.742** |
| Self-Leadership | Leadership Inclusiveness | 0.556 <br> **$-0.003_{ns}$** | NA | 0.556 <br> **$-0.003_{ns}$** |
| Leader Inclusiveness | Psychological Safety | 0.150 <br> **0.394** | NA | 0.150 <br> **0.394** |
| Motivating Language | Role Clarity | $-0.061_{ns}$ <br> **0.284** | 0.219 <br> **0.130** | 0.185 <br> **0.414** |
| Self-Leadership | Role Clarity | 0.294 <br> **0.494** | NA | 0.294 <br> **0.494** |
| Role Clarity | Psychological Safety | *0.114* | NA | *0.114* |

Table Notes: (1) When path differences exist between India and the USA, normal text denotes paths in India and bold text denotes paths in the USA. Paths appear in italics when no significant differences exist between nations. (2) The subscript ns indicates a non-significant path.

**Table 6.** Reduced Model Without Motivating Language.

| From | To | Direct Effects | Indirect Effects | Total Effects |
|---|---|---|---|---|
| Self-Leadership | Trust in Leadership | *0.778* | NA | *0.778* |
| Self-Leadership | Leadership Inclusiveness | *0.590* | NA | *0.590* |
| Self-Leadership | Role Clarity | *0.461* | NA | *0.461* |
| Trust in Leadership | Psychological Safety | *0.184* | NA | *0.184* |
| Leader Inclusiveness | Psychological Safety | 0.142 <br> **0.435** | NA | 0.142 <br> **0.435** |
| Role Clarity | Psychological Safety | *0.115* | NA | *0.115* |
| Self-Leadership | Psychological Safety | $-0.176$ | 0.280 <br> **0.453** | 0.123 <br> **0.296** |

Table Notes: When path differences exist between India and the USA, normal text denotes paths in India and bold text denotes paths in the USA. Paths appear in italics when no significant differences exist between nations.

To provide a quick summary of our results, the Table 7 gives a restatement of our hypotheses and research questions along with our study results.

**Table 7.** Hypotheses and Research Questions, and Model Results.

| Hypothesis/Research Question | Results |
|---|---|
| H$_1$: Motivating language will have a positive relationship with follower psychological safety. | Supported in both nations (total effects) with significant differences between samples |
| H$_2$: Self-leadership will have a positive relationship with follower psychological safety. | Supported in India in the original model and in both nations in the reduced model |
| H$_3$: Leader motivating language will have a positive relationship with follower self-leadership. | Supported in both nations with no differences between nations |
| H$_4$: Motivating language will positively influence a follower's trust in a leader. | Supported in both nations (total effects) with significant differences between samples |
| H$_5$: Self-leadership will positively influence a follower's trust in a leader. | Supported in both nations with significant differences between samples |
| H$_6$: A follower's trust in a leader will positively influence the follower's feeling of psychological safety. | Supported in both nations with no significant differences between samples |
| H$_7$: Motivating language will positively influence a follower's feeling of a leader's inclusiveness. | Supported in both nations (total effects) with significant differences between samples |
| H$_8$: Self-leadership will positively influence a follower's feeling of a leader's inclusiveness. | Supported in Indian in the initial model and both samples in the reduced model with significant differences between nations in both models |
| H$_9$: A follower's feeling of being included by a leader will positively influence the follower's feeling of psychological safety. | Supported in both nations with significant differences between the samples |
| H$_{10}$: Motivating language will positively influence a follower's role clarity. | Supported in both nations (total effects) with significant differences between samples |
| H$_{11}$: Self-leadership will positively influence a follower's role clarity. | Supported in both nations with significant differences between samples |
| H$_{12}$: A follower's role clarity will positively influence the follower's feeling of psychological safety. | Supported in both nations with no significant differences between samples |
| RQ$_1$: Do the constructs of trust in leadership, leadership inclusiveness, and role clarity fully or partially mediate the relationship between leader motivating language and follower psychological safety? | The constructs fully mediate the relationship in both nations |
| RQ$_2$: Do the constructs of trust in leadership, leadership inclusiveness, and role clarity fully or partially mediate the relationship between worker self-leadership and psychological safety? | The constructs partially mediate the relationship in both nations |
| RQ$_3$: Does the model operate in the same way in India and the USA? | The model operates differently in India and the USA, with the effects generally stronger in the USA than in India |

## 3. Discussion

This Study Provided Evidence that Motivating Language and Self-Leadership Have a Positive Influence on a Follower's Feeling of Psychological Safety. In Addition, this Study Was Able to Identify Causal Mechanisms (Trust in Leadership, Leader Inclusiveness, and Role Clarity) by Which This Relationship Operates. Additionally, This Study Showed Significant Differences in Relationships between the Indian and USA Samples

However, the specifics of our findings reveal several intriguing intricacies in these relationships. First, while motivating language had significant links with all three mediating variables in the USA sample, it only had significant indirect links (through a follower's self-leadership) in the Indian sample. This finding contrasts with self-leadership which had significant links with the mediating variables. Additionally, the three mediating variables accounted for the full relationship between motivating language and psychological safety (in both samples), but not between self-leadership and psychological safety.

We can also look at the relationship between motivating language and psychological safety in standard deviation terms, and by comparing this relationship to other relationships in management. By looking at this relationship in terms of standard deviations, we have a useful and common metric to better understand how strong an effect motivating language has on psychological safety. In the Indian sample, for a one standard deviation increase

in motivating language, we expect to see a 0.98 increase in follower psychological safety. In terms of how this relationship compares to other known relationships in management research, it ranks among the top 10% of all relationship strengths (Paterson et al. 2016).

For the USA sample, we can expect that a one standard deviation increase in motivating language use should produce a 1.90 increase in follower psychological safety. In other words, changes in motivating language use lead to an even greater change in follower feelings of psychological safety. In comparison to other known relationships in management research, this link ranks in the top 5% of all such effect sizes.

An interesting and unexpected finding was the overlapping relationship between leader motivating language use and self-leadership. We found that motivating language and self-leadership had a strong enough relationship that including both constructs in the model lead to apparent weak or non-significant relationships between self-leadership and the other exogenous variables. To deal with this issue, we ran a model that did not include motivating language, and found a significant and positive relationship between self-leadership, psychological safety, and all of the mediating variables.

We can also look at the self-leadership → psychological safety link in the same way we examined the motivating language → psychological safety link. In the Indian sample, we can expect a 0.12 standard deviation increase in follower felt psychological safety for every one standard deviation increase in self-leadership. In the USA sample, we can expect a 0.30 increase. While these results indicated smaller links than those of the motivating language construct, they do has a significant and non-negligible effect.

While this finding indicates that both motivating language and self-leadership influence psychological safety, it points to a more interesting result: the role of leader guidance and support in self-leadership application. While most self-leadership research focuses on the individual's role (Neck and Houghton 2006; Stewart et al. 2011), this study underlines the utility of research avenues addressing external influences on self-leadership (Mayfield et al. forthcoming a; Ram 2015). This work points to an intriguing idea about self-leadership—that to promote a follower's psychological well-being, even workers strong in self-leadership need the guidance and support of a leader. This finding should provide practical guidance for leaders: even when a follower can act in self-sufficient way, they still need leadership support.

### 3.1. Implications

Our findings provide insights into how motivating language and self-leadership operate to influence follower psychological safety. This section will explore these ideas for both research and practice.

Study results have strong implications for motivating language's research stream. One research implication comes from showing how leader motivating language has a marked influence on psychological states that help improve a follower's work experience. While prior work has shown motivating language's role in other follower focused outcomes, such as job satisfaction, most motivating language research has focused on outcomes that benefit management and the organization (Mayfield and Mayfield 2018f). As such, we hope this study encourages other researchers to explore how motivating language can contribute to a better workplace.

Findings also indicated that the motivating language → psychological safety link varies between India and the USA, but the reasons for this difference remains unclear. In the Indian sample, motivating language's influence occurs completely through its relationship with self-leadership, while in the USA, it takes a more complicated path through all of the mediating variables. Future research will need to explore why these differences exist.

However, in both nations the mediators fully account for the relationship between leader motivating language use and follower psychological safety. As such, this work provides a strong indication of how the link operates—even if it operates differently in

India and the USA. This finding may inform future research that wishes to explore how motivating language influences other affective states related to psychological safety.

In addition to the research-related possibilities, our finding also have applications for practice. First, leaders should have an awareness of how strongly their communication can influence a follower's psychological safety. While we usually think in terms of how *increases* in motivating language use can *improve* this outcome, leaders must also recognize that drops in motivating language use can have an outsize influence on psychological safety. As such, leaders must strive to continue strong motivating language use after they have improved their communication implementation.

Additionally, leaders can use knowledge of the mediating variables if they do not see improvement in follower psychological safety after increasing their motivating language use. Such a situation may indicate that one or more of the mediators have been blocked by external factors, and thus stopped the expected link. For example, after changes in departmental structure, follower role clarity may diminish. In these cases, leaders must take extra steps to resolve this blockage before they can fully improve follower psychological safety.

For research, however, the self-leadership → psychological safety link provides even more interesting findings than the motivating language link. First, this study indicated a strong overlap between motivating language and self-leadership. This overlap implies that leader behaviors influence self-leadership in ways that have been largely overlooked in the current research stream. This overlap hints that self-leadership needs leadership support in order to improve positive emotional states.

This idea comes from our empirical observation of what happens when you control self-leadership for leadership (communicative) direction, emotional support, and cultural connection—motivating language. With leader motivating language controlled for, self-leadership has no relationship with psychological safety in the USA—in contrast, the relationship without such a control was positive and stronger in both nations.

*3.2. Limitations and Future Research*

We also need to address study limitations and where researchers might expand on this work. For limitations, we must acknowledge the correlational—rather than causal—nature of this study. The study design does not permit full causal testing, but does permit an initial test of this idea through examining model linkages. However, future work should employ such designs as experiments or time-lagged examinations to address this limitation.

Similarly, study findings were limited in generalizability. The Indian and USA samples drew on a broad cross-section of participants, but future researchers might want to examine the relationships in specific demographic groups to test for moderating effects. In addition, given the differences in models between nations, we must be cautious in drawing conclusions about how the model operates across nations. Findings did indicate similar support for the general link relationships between motivating language, self-leadership, and psychological safety. However, the strength of these relationships differed, indicating some national effect in operation. As such, future work should examine the model in different nations, and—when possible—measure national characteristics for testing purposes. In this way, researchers can examine how such national characteristics influence model differences.

This study was also limited by its utilizing a mono-method. While a survey based, correlational study can provide many insights into a phenomenon, multiple methods can provide a more faceted understanding. Future research should combine such diverse methods as qualitative research, experimental design, and even simulation studies (such as agent based modeling).

While future studies need to fully explore study results, we can think through some of the possible implications now. By controlling for leader motivating language use, the self-leadership construct in the full model artificially represents what self-leadership would look like if someone had none of the benefits of a leader's motivating language. From the

model findings, it seems that—in the USA—a person's self-leadership does not offer a buffer against negative workplace situations. In short, a person may have the ability to set their own goals and work towards them, but without leadership support, this ability gives no feeling of security—they may be doing the wrong things for the wrong reasons.

Another intriguing aspect of this finding comes from the differences between the nations. The Indian sample did show a significant and positive relationship between self-leadership and psychological safety for the full model. As such, it appears that leader motivating language plays a different—and perhaps smaller—role in India than in the USA. This finding goes along with the differences in how strongly self-leadership operates between India and the USA. In the reduced model, the USA sample showed a stronger relationship between self-leadership and psychological safety than in India. Future research may look for possible national differences as moderators for why these variances exist.

As other avenues for future research, our results showed that the study mediators fully accounted for how motivating language influences psychological safety. This finding indicates that we have a good understanding of how motivating language operates in relationship to psychological safety.

However, the mediators only partially accounted for the relationship between self-leadership and psychological safety. In itself, this finding is not that interesting—we have many relationships that we only partly understand. What makes it more intriguing is that the non-mediated relationship (the direct path between self-leadership and psychological safety) presents as negative. This indicates that whatever hidden mechanism exists results in stronger self-leadership being a detriment to psychological safety, and gives future researchers an interesting place to examine both constructs.

Exploring this negative link can add a better understanding of how self-leadership operates. The finding hints at situations where self-leadership may create negative outcomes for the follower. While exploring such a link goes beyond this study's scope, future researchers may want to explore this idea.

For practice, as with motivating language, the link between self-leadership and psychological safety indicates the potential it holds for improving the workplace. Leaders should think about ways they can encourage follower use of self-leadership, such as training or using motivating language to enhance these behaviors. Results also indicate that all workers have a self-interest in improving their self-leadership skills so that they can feel more secure in their workplace.

In addition, more work needs to look at this model—and motivating language and self-leadership in general—across a wider range of nations. While recent years have seen an increase in global research on both constructs (Mayfield and Mayfield 2018f; Ram 2015), we still only have limited work in European, African, and Middle Eastern nations. To truly understand the effects of these constructs, we need to have a studies in a wider variety of cultures.

**Author Contributions:** Conceptualization, M.M. and J.M.; writing, M.M. and J.M.; methodology, M.M. All authors have read and agreed to the published version of the manuscript.

**Funding:** This research was funded by the A. R. Sanchez Jr. School of Business, Texas A&M International University.

**Institutional Review Board Statement:** The study was conducted according to the guidelines of the Declaration of Helsinki, and approved by the Institutional Review Board of Texas A&M International University, May 2020.

**Informed Consent Statement:** Informed consent was obtained from all subjects involved in the study.

**Data Availability Statement:** Interested parties can obtain the study data by contacting the article authors.

**Conflicts of Interest:** The authors declare no conflict of interest.

**Appendix A. Selected Study Scales**

*Appendix A.1. Motivating Language Scale*

Originally created by Jacqueline and Milton Mayfield.

The examples below show different ways that your boss might talk to you. Please use the following selections to choose the answer that best matches your perceptions, and then click on the appropriate response.

(All items had response choices as follows: Very Little, A Little, Some, A Lot, A Whole Lot)

Direction-Giving Language

1. Gives me useful explanations of what needs to be done in my work.
2. Offers me helpful directions on how to do my job.
3. Provides me with easily understandable instructions about my work.
4. Offers me helpful advice on how to improve my work.
5. Gives me good definitions of what I must do in order to receive rewards.
6. Gives me clear instructions about solving job-related problems.
7. Offers me specific information on how I am evaluated.
8. Provides me with helpful information about forthcoming changes affecting my work.
9. Provides me with helpful information about past changes affecting my work.
10. Shares news with me about organizational achievements and financial status.

Empathetic Language

11. Gives me praise for my good work.
12. Shows me encouragement for my work efforts.
13. Shows concern about my job satisfaction.
14. Expresses his/her support for my professional development.
15. Asks me about my professional well being.
16. Shows trust in me.

Meaning-Making Language

17. Tells me stories about key events in the organization's past.
18. Gives me useful information that I couldn't get through official channels.
19. Tells me stories about people who are admired in my organization.
20. Tells me stories about people who have worked hard in this organization.
21. Offers me advice about how to behave at the organization's social gatherings.
22. Offers me advice about how to "fit in" with other members of this organization.
23. Tells me stories about people who have been rewarded by this organization.
24. Tells me stories about people who have left this organization.

The original authors released this scale under a Creative Commons Attribution-ShareAlike 4.0 International license. You can find information on this license at the following site: https://creativecommons.org/licenses/by-sa/4.0/ (accessed on 1 April 2021).

*Appendix A.2. Comparative Taste Preference (Common Methods Bias Marker) Scale*

By Milton and Jacqueline Mayfield.

Instructions: Please read each of the following statements and select how strongly you agree or disagree with it. (Response format 1 to 5, Strongly Agree to Strongly Disagree.)

I like my food spicier than other people do.

My food can never be too spicy.

I add more spices to my food than anyone I know.

I choose spicier dishes when eating out than anyone else.

I prefer mild food.

The original authors released this scale under a Creative Commons Attribution-ShareAlike 4.0 International license. You can find information on this license at the following site: https://creativecommons.org/licenses/by-sa/4.0/ (accessed on 1 April 2021).

The following scales have not been released under licenses permitting full reproduction, so we have provided item examples and response scales

*Appendix A.3. Self-Leadership*

Response Scale: Strongly Disagree, Disagree, Neither Agree nor Disagree, Agree, Strongly Agree
Example Items:
I establish specific goals for my own performance
Sometimes I picture in my mind a successful performance before I actually do a task
I think about my own beliefs and assumptions whenever I encounter a difficult situation

*Appendix A.4. Organizational Trust Scale*

Response Scale: Nearly Zero, Very Low, Low, 50-50, High, Very High, Near 100%
Example Items:
My level of confidence that my supervisor is technically competent at the critical elements of his or her job is:
My level of confidence that my supervisor has an acceptable level of understanding of his/her job is:
My level of confidence that my supervisor will think through what he or she is doing on the job is:

*Appendix A.5. Inclusive Leadership*

Response Scale: Not At All, A Little, Sometimes, To a Limited Extent, To a Large Extent
Example Items
The manager is open to hearing new ideas
The manager is available for consultation on problems
The manager encourages me to access him/her on emerging issues

*Appendix A.6. Role Clarity (Role Ambiguity Scale)*

Response Scale: Strongly Disagree, Disagree, Neutral, Agree, Strongly Agree
Example Items
I know exactly what is expected of me in my job.
I feel certain about the level of authority I have.
I know what my responsibilities are.

*Appendix A.7. Social Desirability Scale*

Response Scale: True, False
Example Items:
It is sometimes hard for me to go on with my work if I am not encouraged.
I am always courteous, even to people who are disagreeable.
I have never deliberately said something that hurt someone's feelings.

*Appendix A.8. Psychological Safety (with boss)*

Response Scale: Completely Disagree, Somewhat Disagree, Neutral, Somewhat Agree, Completely Agree
Example Items:
If you make a mistake, your boss will hold it against you.
It is safe to take a risk with your boss.
Your boss values your unique skills and talents.

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
