# Peer review of "Sound and Safe: The Role of Leader Motivating Language and Follower Self-Leadership in Feelings of Psychological Safety"

_admsci, doi:10.3390/admsci11020051_

Round 1

Reviewer 1 Report

The undertaken issue is interesting from the research point of view. The scope of research hypotheses and questions is far too broad. Hypotheses are inconsistent. The answers to most of the research questions are obvious. The research project is similar to studying the effects of sleep on health condition. Moreover, the study was conducted on a random group of people. There is no information about the research sample and methods of its determination. There is no justification why the people from the USA and India were hit. The scientific value of the software is nil - ZERO.

Author Response

Reviewer 1

Comment: Need information on the research sample and its determination

Response: Thank you for pointing out that readers may want more information on why we selected the two nations for the study. We included a paragraph detailing why we selected our subject in the paper’s sample section. This new paragraph is as follows:

We selected these nations for the insights we hoped they could provide into motivating language and self-leadership. The USA has seen substantial research on the two focal constructs (Mayfield 2018, Neck & Houghton, 2006), and a sample from this nation provides a base to compare results from other nations. In contrast, researchers have conducted fewer investigations on either construct in India, but this nation provides a good potential contrast for findings from the USA. India possesses a culture with markedly different cultural characteristics, demographic aspects, and even predominate religious beliefs from the USA. Therefore, findings of similar results across these two nations provide good evidence of the model’s generalizability, while differences can provide avenues for future researchers to explore to try an identify why the differences exist (i.e. which national characteristics create the differences).

Comment: “The scientific value of the software is nil - ZERO.”

Response: According to an analysis by Muenchen (2012), R was the second most cited statistical software in academic publications (after SPSS and ahead of SAS). Similarly, a Google Scholar search turns up over 17,000 citations for lavaan. Both of these software packages appear to have a strong presence and utility in academic research. We clarified the academic status of the software in the following paragraph.

We selected lavaan and R for our analysis software because of their flexibility, substantial use in the research community, and the quality assurance benefits attendant with open source software (Muenchen, 2012; Zhao & Elbaum, 2003). With its integration into R, researchers can seamlessly combine data testing (as for reliability), manipulations (as with the removing bias stage in this research), and visualizations (as with the correlations matrices). This workflow ease allows for greater focus on interpreting the results. In addition, R software has become the second most cited software among academic scholars (Muenchen, 2012).

Comment: “There is no justification why the people from the USA and India were hit.”

Response: We think there was a misunderstanding about what “hit” means. It is a term used by Amazon for recruiting subjects. No subjects were hit in the study. The respondents completed an online survey at a place of their choosing. No physical contact occurred between the researchers and the respondents.

Comment: Hypotheses too broad

Response: We do not know how to address this comment. Each hypothesis has the form of Construct A has a positive influence on Construct B. As such, the hypotheses breaks down the overall model into its smallest constituent parts, and even posits a direction in which we expect the relationship to act.

To make the hypotheses more specific, we would have to make predictions about the relationship strengths and mathematical form, and not enough information about the relationships exist to make such statements.

The research questions are somewhat broad, but this broadness comes from a desire to explore aspects of the model for which no strong theoretical or empirical knowledge exist. As such, using more limited statements as research questions seems counterproductive.

Comment: Hypotheses inconsistent

Response: Again, we are not sure about how to address this comment. The hypotheses follow the same format throughout the manuscript with only small variations necessary for grammatical correctness or to improve hypothesis readability.

Comment: “The answers to most of the research questions are obvious. The research project is similar to studying the effects of sleep on health condition.”

Response: While we broadly agree that most people will expect positive leadership and individual behaviors to lead to positive follower outcomes, we disagree with the specific statement. The study examined two specific antecedent behaviors – leader motivating language and follower self-leadership, their effect on a specific follower outcome (psychological safety), and specific mediating mechanisms by which this relationship operates. As such, a better health study analogy seems to be how sleep apnea and shift work influences alertness as mediated by blood oxygen and cortisol levels.

In addition, the study did turn up unexpected findings in the relationship between motivating language and self-leadership, and the mediation process. As such, not all of the results seem to be obvious. From these unexpected results, we added paths for future research in the discussion.

Reviewer 2 Report

First of all, thank you for the opportunity to review this work. It is an interesting manuscript, especially due to the novelty of the variables analyzed and the model it proposes. psychological safety is a current variable and it needs models that improve its knowledge.

The manuscript is drafted with great clarity and the results are also clearly stated.

The improvements that are required are mainly in appointments. I was surprised by the large number of citations that include the initials of the authors and no initials. I do not know if there is any reason to put some quotes with their initials and others without the initials, but I think it enables an explanation by the authors.

The other improvement has to do with the description of the tools. I think a greater description of the measurement tools is adequate. This should include examples of the items for all of them, response scales, etc.

However, and despite these limitations, I congratulate the authors for the great work. 

Author Response

Reviewer 2

Comment: Problem with/explanation of citations – some appear with initials and others without.

Response: We understand how these citations look strange. They look strange to us as well. However, we checked, and all of the citations seem to follow the current APA format citation guidelines. APA uses initials (and full first names) to help distinguish between different authors with the same last name (such as J. Mayfield & M. Mayfield), when the same author’s name appears in different contexts (such as A.C. Edmondson) and even has guidelines depending on how often the author names appear and in what context. The rules about how to present an author’s name and which situations can become quite complex, and we – frankly – use a citation management software to help us keep track.

Comment: The other improvement has to do with the description of the tools. I think a greater description of the measurement tools is adequate. This should include examples of the items for all of them, response scales, etc.

Response: Thank you for this suggestion. Including example items will help the reader get a better sense of what was actually measured.

We could include the full versions of the Motivating Language scale, and the Comparative Taste Preference scale in Appendix A because those scales have been released under a Creative Commons license. We included example items from the other scales in the same appendix, and the response scales for all of the surveys as well.

Reviewer 3 Report

Figure 2 and Figure 3 are not well understood. What are the meanings of the vertical and horizontal axes in each of these figures? They are often used in psychology, but it is difficult for specialists in unfamiliar fields to grasp them intuitively.

What is the difference between the lavaan package used in the analysis and PLS or AMOS? Why was PLS or AMOS not used to analyse this sample? The differences in the study should not be a black box and should be fully explained.

For the model in Figure 1, the indicators for constructing each variable are not shown. What sub-variables make up each of the circled variables? Please add a detailed explanation.

A fatal aspect of this paper is the absence of a concluding chapter. It ends with only a discussion, and the interpretation of the analysis is not fully stated. In short, it is unclear how the findings of this paper are novel to previous studies, and the practical implications are not drawn out.

Finally, it is advisable to prepare and resubmit a "Discussion" as an interpretation of the analysis and a "Conclusion" that summarises the study's findings.

Author Response

Reviewer 3

Comment: Explain the horizontal and vertical axes of figures 1 & 2

Response: Thank you for this suggestion. We too often become so used to a presentation method that we forget it can be unfamiliar to others. We added the following to figures 1 and 2 (respectively)

The numbers along the row or column segments show the items range. For example, the segment in the upper left corner shows that the self-leadership scores ranges from 1 to 5.”

The numbers along the row or column segments show the items range after controlling for bias. These numbers generally follow a normal distribution, but some outliers remain, thus creating non-symmetrical ranges. As an example, the trust in supervisor variable (right hand column, second row) shows that most observations have a value between -2 and 2, but some outliers push the lower range to -4.”

Comment: Explain why the study used lavaan instead of PLS or Amos. Give an explanation in the text.

Response: Thanks for pointing this issue out as something that might interest readers. While analysis software choice will always rest – to some extent – on an author’s personal preferences, we did select the software based on its quality, flexibility, and support in the academic community. We discussed the qualities of R/lavaan, but did not directly mention WarpPLS or AMOS, because we know that these software packages provide quality results as well and we did not want to imply otherwise. We included the following text in the manuscript after the first paragraph in the Results section:

We selected lavaan and R for our analysis software because of their flexibility, substantial use in the research community, and the quality assurance benefits attendant with open source software (Muenchen, 2012; Zhao & Elbaum, 2003). With its integration into R, researchers can seamlessly integrate data testing (as for reliability), manipulations (as with the removing bias stage in this research), and visualizations (as with the correlations matrices). This workflow ease allows for greater focus on interpreting the results. In addition, R software has become the second most cited software among academic scholars (Muenchen, 2012).

Comment: Give a list of indicators used for each construct.

Response: Thank you for the good suggestion. It makes sense to let the reader know how the model variables were created in a prominent place such as the figure. We updated the figure text to describe how we created each construct. We were not able to include the indicants in the figure without making the overall figure too small to easily read. We hope that the compromise of giving a narrative about our process addresses your concern.

Comment: Provide a Conclusion section that summarizes the study findings and a Discussion section that interprets/explores the results.

Response: Thanks for this suggestion. It makes sense to split our (previous) Conclusion section into two parts, each focusing on a different aspect of what we want the reader to take away from our work. We have done so by providing a Conclusion section that provides more details on our analysis results, and a Discussion section that goes into more details about how our work expands the current literature and how people can use our findings in applied settings. We will provide more details on the Discussion section in our next two responses.

For the Conclusion section, we added the following paragraphs:

We can also look at the relationship between motivating language and psychological safety in standard deviation terms, and by comparing this relationship to other relationships in management. By looking at this relationship in terms of standard deviations, we have a useful and common metric to better understand how strong an effect motivating language has on psychological safety. In the Indian sample, for a one standard deviation increase in motivating language, we expect to see a 0.98 increase in follower psychological safety. In terms of how this relationship compares to other know relationship in management research, it ranks among the top 10% of all relationship strengths (Paterson, et al., 2016).

For the USA sample, we can expect that a one standard deviation increase in motivating language use should produce a 1.90 increase in follower psychological safety. In other words, changes in motivating language use lead to an even greater change in follower feelings of psychological safety. In comparison to other known relationships in management research, this link ranks in the top 5% of all such effect sizes.

We can also look at the self-leadership → psychological safety link in the same way we examined the motivating language → psychological safety link. In the Indian sample, we can expect a 0.12 standard deviation increase in follower felt psychological safety for every one standard deviation increase in self-leadership. In the USA sample, we can expect a 0.30 increase. While these results indicated smaller links than those of the motivating language construct, they do has a significant and non-negligible effect.

Comment: Include a Discussion section fully exploring how the work extends the current literature.

Response: In the new Discussion section, we added an exploration of what our findings might mean for research. The paragraphs are as follows:

Our findings provide insights into how motivating language and self-leadership operate to influence follower psychological safety. This section will explore these ideas for both research and practice.

Our findings have strong implications for motivating language’s research stream. One research implication comes from showing how leader motivating language has a strong influence on psychological states that help improve a follower’s work experience. While prior work has shown motivating language’s role in other follower focused outcomes – such as job satisfaction – most motivating language research has focused on outcomes that benefit management and the organization (J. Mayfield & Mayfield, 2018f). As such, we hope this study encourages other researchers to explore how motivating language can contribute to a better workplace.

Findings also indicated that the motivating language → psychological safety link varies between India and the USA, but the reasons for this difference remains unclear. In the Indian sample, motivating language’s influence occurs completely through its influence on self-leadership, while in the USA, it takes a more complicated path through all of the mediating variables. Future research will need to explore why these differences exist.

However, in both nations the mediators fully account for the relationship between leader motivating language use and follower psychological safety. As such, this work provides a strong indication of how the link operates – even if it operates differently in India and the USA. This finding may inform future research that wishes to explore how motivating language influences other affective states related to psychological safety.

. . .

For research, however, the self-leadership → psychological safety link provides even more interesting findings than the motivating language link. First, this study indicated a strong overlap between motivating language and self-leadership. This overlap implies that leader behaviors influence self-leadership in ways that have been largely overlooked in the current research stream. This overlap hints that self-leadership needs leadership support in order to improve positive emotional states.

. . .

Exploring this negative link can add a better understanding of how self-leadership operates. The finding hints at situations where self-leadership may create negative situations for the follower. While exploring such a link goes beyond this study’s scope, future researchers may want to explore this idea.

Comment: Include a Discussion section fully exploring the findings’ practical implications.

Response: In the new Discussion section, we added an exploration of what our findings might mean for practice. The paragraphs are as follows:

In addition to the research related possibilities, our finding also have applications for practice. First, leaders should have an awareness of how strongly their communication can influence a follower’s psychological safety. While we usually think in terms of how increases in motivating language use can improve this outcome, leaders must also recognize that drops in motivating language use can have an outsize influence on psychological safety. As such, leaders must strive to continue strong motivating language use after they have improved their communication implementation.

Also, leaders can use knowledge of the mediating variables if they do not see improvement in follower psychological safety after increasing their motivating language use. Such a situation may indicate that one or more of the mediators have been blocked by external factors, and thus stopped the expected link. For example, after changes in departmental structure, follower role clarity may diminish. In such cases, leaders must take extra steps to resolve this blockage before they can fully improve follower psychological safety.

. . .

For practice, as with motivating language, the link between self-leadership and psychological safety indicates the potential it holds for improving the workplace. Leaders should think about ways they can encourage follower use of self-leadership – such as training or using motivating language to encourage these behaviors. Results also indicate that all workers have a self-interest in improving their self-leadership skills so that they can feel more secure in their workplace.

Round 2

Reviewer 1 Report

Research results have only limited cognitive value. The results have no scientific value. The authors did not select the research sample. The method to justify sending the questionnaire to 400 Indian and 400 US residents was not used. This type of manuscript does not qualify for publication in a scientific journal.

Author Response

Thank you for your time and effort in providing feedback. Unfortunately, it seems we have reached an impasse. While we have tried to address your concerns, it appears that they rest on a fundamental disagreement with our research model. Since we have followed a model that the vast majority of social science work uses, we do not know what we can do to further address your concerns. Without more specifics on these concerns, we cannot even provide our rationale for why we made the research/analysis choices we used.

Reviewer 3 Report

This is the first suggestion to put discussion after the conclusion. I think it's a very innovative and bold proposal, but I'm too old-fashioned to accept it.

Author Response

Thank you for your feedback and honesty in your review. We have re-arranged and expanded the discussion/conclusion section to follow your suggestion.